# Personnel Well-Being in the Helsinki University Hospital during the COVID-19 Pandemic—A Prospective Cohort Study

**DOI:** 10.3390/ijerph17217905

**Published:** 2020-10-28

**Authors:** Henna Haravuori, Kristiina Junttila, Toni Haapa, Katinka Tuisku, Anne Kujala, Tom Rosenström, Jaana Suvisaari, Eero Pukkala, Tanja Laukkala, Pekka Jylhä

**Affiliations:** 1Department of Psychiatry, University of Helsinki and HUS Helsinki University Hospital, and Finnish Institute for Health and Welfare, 00029 HUS Helsinki, Finland; henna.haravuori@thl.fi; 2HUS Helsinki University Hospital, Nursing Research Center and University of Helsinki, 00029 HUS Helsinki, Finland; kristiina.junttila@hus.fi (K.J.); toni.haapa@hus.fi (T.H.); 3Department of Psychiatry, University of Helsinki and Acute Psychiatry and Consultations, HUS Helsinki University Hospital, 00029 HUS Helsinki, Finland; katinka.tuisku@hus.fi (K.T.); pekka.jylha@hus.fi (P.J.); 4HUS Helsinki University Hospital and University of Helsinki, 00029 HUS Helsinki, Finland; anne.kujala@hus.fi; 5Department of Psychology and Logopedics, Faculty of Medicine, University of Helsinki, 00014 Helsinki, Finland; tom.rosenstrom@helsinki.fi; 6Department of Public Health Solutions, Finnish Institute for Health and Welfare, Mental Health Unit, 00271 Helsinki, Finland; jaana.suvisaari@thl.fi; 7Faculty of Social Sciences, Tampere University, 33100 Tampere, Finland; eero.pukkala@cancer.fi

**Keywords:** COVID-19 pandemic, Finland, healthcare personnel, psychological distress, post-traumatic stress disorder

## Abstract

In March 2020, strict measures took place in Finland to limit the COVID-19 pandemic. Majority of Finnish COVID-19 patients have been located in southern Finland and consequently cared for at the Hospital District of Helsinki and Uusimaa (HUS) Helsinki University Hospital. During the pandemic, HUS personnel’s psychological symptoms are followed via an electronic survey, which also delivers information on psychosocial support services. In June 2020, the baseline survey was sent to 25,494 HUS employees, 4804 (19%) of whom answered; altogether, 62.4% of the respondents were nursing staff and 8.9% were medical doctors. While the follow-up continues for a year and a half, this report shares the sociodemographic characteristics of the respondents and the first results of psychological symptoms from our baseline survey. Out of those who were directly involved in the pandemic patient care, 43.4% reported potentially traumatic COVID-19 pandemic-related events (PTEs) vs. 21.8% among the others (*p* < 0.001). While over a half of the personnel were asymptomatic, a group of respondents reported PTEs and concurrent depression, insomnia, and anxiety symptoms. This highlights the need to ensure appropriate psychosocial support services to all traumatized personnel; especially, nursing staff may require attention.

## 1. Introduction

COVID-19 (severe acute respiratory syndrome coronavirus 2, SARS-CoV-2) outbreak began in December 2019 in Wuhan, China, and caused a pandemic with over 41.0 million confirmed infections and over 1.1 million deaths by 21 October 2020 [1]. In most cases, the virus causes only a mild disease. The severe and possibly life-threatening complications of the infection include acute lung injury, acute respiratory distress syndrome, and multiple organ failure [2,3]. In Finland, from the global perspective, strict restrictions have taken place to slow down the pandemic by preventing physical contact between people. This is of utmost importance to secure intensive care (IC) capacity to those with severe symptoms. Majority of COVID-19 pandemic patients in Finland have been cared for at the HUS Helsinki University Hospital since March 2020. By 21 October 2020, there is detailed information on 352 deaths caused by the COVID-19 pandemic in Finland; 48% and 52% of them were males and females, respectively, with a median age of 84 years. Before they died of COVID-19, 35% were cared for by the primary healthcare, 20% by special medical care, 43% by social welfare services, 2% at home or elsewhere (see also www.thl.fi/en).

Healthcare personnel face unique challenges during the COVID-19 pandemic. In China, the Wuhan area, frontline nurses and doctors caring for COVID-19 patients reported an increase in symptoms of depression, insomnia, anxiety, and psychological distress as compared to other healthcare personnel [4,5]. The first general population studies from the Wuhan area report similar findings with somewhat lower symptom intensities [6,7]. In Europe, a study of 110 nurses and doctors from Germany reported that the nurses working in COVID-19 wards are especially affected psychologically [8]. Work-related stress, long work shifts, and contagion were a concern in Italy [9]. Research in this field is limited and, according to our knowledge, only a few studies from Europe are currently available [8,10,11].

The basic principles of high-quality psychosocial support [12,13,14,15,16,17,18,19] have emerged from several international reports assessing the immediate needs of the healthcare personnel caring for COVID-19 patients can shortly be summarized as follows: Listen, Supply, Prepare, Support, and, if needed, Care for us and our close ones [14,15,16,17,18,19]. Timing is of importance in assessing stress-related symptoms; assessment before at least one month since a potentially traumatic event is prone to wrong positive findings [20]. The COVID-19 pandemic increases the risk of exposure to potentially traumatic events among healthcare personnel in professional and private life, while the pandemic itself is not always a traumatic event to everyone exposed to it [12,13].

In this study, the personnel’s well-being at the HUS Helsinki University Hospital during the COVID-19 pandemic is followed via an electronic survey. We report the baseline results from the prospective cohort study on the HUS personnel’s psychological symptoms conducted in June 2020.

## 2. Materials and Methods

This report shares the first baseline results of an ongoing prospective HUS personnel well-being cohort study (HUS HEHY COVID-19) in the southern Finland area. This study was approved by the HUS Ethics Committee and the permission to conduct the study was obtained from the Joint Authority of the Helsinki and Uusimaa Hospital District. An electronic survey was created to assess the well-being of the HUS personnel. It consists of sociodemographic background questions and five symptom rating scales: Mental Health Index (MHI-5), Insomnia Severity Index (ISI), Patient Health Questionnaire-2 (PHQ-2, also referred to as PRIME-MD), Primary Care Post-Traumatic Stress Disorder Scale (PC-PTSD-5), and Overall Anxiety and Impairment Scale (OASIS). These scales assess psychological distress, insomnia, depression symptoms, traumatic experiences (with questions focused on work-related experiences with COVID-19 patients), trauma-related psychological symptoms, and anxiety [21,22,23,24,25,26,27]. In addition, the survey includes questions about potential changes in respondents’ daily work and their adjustment to the changes, respondents’ attitudes towards COVID-19 patients, and a few open questions about their need for psychosocial support. The survey was delivered in Finnish and Swedish (the major languages of the HUS personnel). The survey took about 10–15 min to answer. Initially, all the employees with a functional HUS e-mail address (N = 25,494) were invited to participate in the baseline survey. Due to possible personnel work changes and turnover, an open access link is also available on the HUS personnel’s internal website (HUS Intra). The majority of answers were received through the e-mail survey when launching the study during the period from 4 June to 26 June 2020, but we also included the results from the open access link from the same timeframe, compared the answerers, and reported possible differences.

IBM SPSS Statistics for Windows, Version 26.0 (Armonk, NY, USA: IBM Corp.) and R version 4.0.2 (2020-06-22) were used in the statistical analyses. We examined 2-way tables and chi-squared tests in the former and multivariate (multiple) logistic regression models in the latter to evaluate interaction effects of a COVID-19 contact, potentially traumatic work-related events (PTEs), and the nursing staff membership on the psychological outcomes, as well as to adjust the main effects for each other.

## 3. Results

Table 1 describes the sociodemographic background of 4804 HUS employees (19% of the HUS personnel) who participated in the June electronic survey. Pandemic-related changes at work, different potentially traumatic work-related events (PTEs), and the MHI-5, ISI, PHQ-2, and PC-PTSD-5 results from the whole sample are reported in Table 2. PTEs at work were also more common among the nursing staff as compared to other respondents (34.6%; *n* = 1011 vs. 16.5%; *n* = 284).

Table 3 describes differences between the personnel directly caring for COVID-19 pandemic patients vs. the other personnel. Briefly, there was a statistically significant difference between the frontline and the other personnel in psychological distress (MHI-5), insomnia (ISI), and depression symptoms (PRIME-MD). Potentially traumatic events related to the COVID-19 pandemic were more common among the personnel directly in contact with pandemic patients. PC-PTSD-5 scale recognized an almost equal proportion of respondents in both groups, with 23–24% having a high risk of PTSD.

In addition, we evaluated whether different rating scales recognized the same respondents at higher risk of psychiatric comorbidity. Four groups emerged, where 54.3% had no self-reported symptoms (N = 2463), 17.9% had psychological symptoms without pandemic work-related traumatic events (N = 811), 14.6% (N = 664) reported pandemic work-related traumatic events and also depression, insomnia, and anxiety symptoms, and 13.2% (N = 598) had pandemic work-related traumatic events without depression symptoms, but with some symptoms of anxiety or stress.

Table 4 reveals that potentially traumatic COVID-19 pandemic work-related events strongly predicted psychological distress indexed by MHI-5 (model 2). Both belonging to the nursing staff and participating in direct care of COVID-19 patients were independently associated with psychological distress (model 1, Table 4), and these associations were explained by (i.e., lost significance upon adjusting for) experiencing COVID-19 pandemic work-related traumatic events (model 2). Furthermore, participation in direct care of COVID-19 patients was not associated with psychological distress in the non-nursing staff (model 3), though relatively few non-nurses participated (n = 136 vs. *n* = 1094). Age, gender, or working as nurse did not predict who of the respondents with PTEs developed PTSD symptoms (data not shown). The e-mail respondents (N = 4614) were compared with the HUS Intra open access link respondents (N = 190), and they answered five days later (M = 8.6 vs. M = 13.6 days). The open link answerers were also slightly younger (M = 44.3 vs. M = 41.9 years).

## 4. Discussion

The HUS personnel in direct contact with COVID-19 patient care reported more psychological distress than other personnel in the June 2020 baseline survey. Potentially traumatic experiences related to the COVID-19 pandemic were of significance among all personnel. However, it is important to note that these data consist of self-reported symptoms and the respondents represent a selected group of the HUS personnel (19%). Those who took time to respond may have been more involved with the COVID-19 pandemic. For comparison, prevalence of the psychosocial burden on nurses or other health professionals regarding depression and anxiety has been reported to be 22.8% and 23.2%, respectively. Insomnia prevalence has been estimated to be 38.9% [28].

Clinically significant psychological distress in the Finnish population measured with the MHI-5 using the same cut-off score as in the current study is monitored by the the FinSote national survey of health, well-being, and service use (see https://thl.fi/en/web/thlfi-en/research-and-expertwork/population-studies/national-finsote-survey). In the most recent survey conducted in years 2017–2018, the prevalence of psychological distress in the age group of 20–54 years was 13.3% in men and 14.8% in women, and in the age group of 55–74 years, the prevalence was 8.4% in men and 7.9% in women (data available online at http://www.terveytemme.fi/finsote/alueet2018/terveys.html). Compared to these figures, the prevalence of psychological distress was higher in the current study, particularly among women. The high level of psychological distress is consistent with the results from other mental health scales. Of note, there is no universally accepted MHI-5 cut-off score for clinically significant psychological distress. The cut-off score used in this study indicates a symptom severity where some mood or anxiety disorder is quite likely [29].

The prevalence of insomnia symptoms in the working age population in Finland is 9.2–9.6% [30] corresponding to the insomnia rates in the non-exposed employees of our sample (9.4%). The employees who directly worked with COVID-19 patients showed instead a significantly higher rate (12.3%) of clinical insomnia symptoms. Among the Finnish employees, insomnia symptoms are associated with a subsequent risk of sickness absence [31].

According to DSM-5, an etiological traumatic event for post-traumatic stress disorder is defined as follows: the person is exposed to death, threat of death, actual or threat of a serious injury, or actual or threat of sexual violence by direct exposure, witnessing a trauma, learning that a relative or close friend was exposed to a trauma or indirect exposure to aversive details of a trauma in the course of professional duties [20]. Most people (2/3) recover from traumatic events with social support from close ones [12,13]. Prolonged exposure and earlier individual vulnerability, including earlier trauma exposure, especially to several traumas, are risk factors of later stress-related symptoms, which, after prolonged exposure such as a pandemic, may affect 25–30% of those at risk [13]. In this study, 23% of the respondents with pandemic-related PTEs reported PTSD symptoms, and exposure to pandemic-related PTEs predicted psychological distress.

Studies from China have found that the frontline healthcare personnel, especially the nursing staff, caring for patients with COVID-19 are at risk of anxiety and mental health problems [4,32]. Similar results have been described in studies from Germany [8], Israel [33], Portugal [34], and Turkey [35]. Moreover, it has been identified that the perceived threat of COVID-19 enhances turnover intentions among nurses [36]. In this study, nurses also appeared to suffer a heavier psychological load from treating COVID-19 patients than other professionals. There is a demand for stronger psychosocial or psychotherapeutic support, especially for nurses, and the already existing support possibilities such as peer and team support could be used [37].

## 5. Conclusions

The studies regarding the well-being of healthcare personnel during the COVID-19 pandemic have emphasized the need to provide psychosocial support for the frontline personnel [31,34,38]. Caring for children and young ones that remind of one’s own children or incidentally caring for close ones or older relatives may cause distress even to experienced healthcare personnel, who otherwise may be more challenged by the amount of work during the pandemic than by psychological exposure to disease and death. In Finland, in addition to the frontline personnel, especially the nursing staff, all the personnel who report potentially traumatic events related to the COVID-19 pandemic work require attention and support.

## Figures and Tables

**Table 1 ijerph-17-07905-t001:** Sociodemographic background information of the HUS personnel participants of the well-being survey.

Sociodemographic Variable	Whole Sample *N* = 4804 ^1^	
Age, *n* = 4494, Mdn = 45, M (SD)	44.2	(11.4)
Gender, *n* (%)		
male	538	(11.4)
female	4130	(87.5)
other or prefer not to answer	51	(1.1)
Highest education, *n* (%)		
primary and lower secondary education	75	(1.6)
upper secondary education	773	(16.3)
Bachelor’s or equivalent	2605	(54.9)
Master’s or equivalent	797	(16.8)
Doctoral or equivalent	488	(10.3)
other	5	(0.1)
Personnel group, *n* (%)		
nursing staff	2989	(62.4)
medical doctors	425	(8.9)
special personnel (including psychologists and social workers)	377	(7.9)
other (non-healthcare) personnel	1001	(20.9)

^1^ Initially 4840, 36 duplicate answers removed.

**Table 2 ijerph-17-07905-t002:** Potentially traumatic events related to work with COVID-19 patients, work changes, and psychological symptoms among the HUS personnel participants of the well-being survey.

Self-Reported Changes in the Work and Psychological Distress Symptoms in the June 2020 Survey	Whole Sample *N* = 4804	
Changes in work due to COVID-19, *n* (%)		
yes	3943	(82.4)
no	844	(17.6)
Were you in contact with confirmed or suspected cases of COVID-19 last week? *n* (%)		
directly cared for	1227	(25.6)
other answers	3560	(74.4)
Did you feel **a need for psychological support** last month? *n* (%)		
yes	774	(16.3)
no	3966	(83.7)
Did you **receive support** through a well-being project for the personnel, from occupational healthcare, or otherwise through the HUS employer organization last month? *n* (%)		
yes	397	(8.4)
no	4332	(91.6)
Mental Health Index, MHI-5, *n* (%)		
>52, no psychological distress	3975	(83.3)
≤52, psychological distress	797	(16.7)
N = 4772, Mdn = 76, M (SD)	73.3	(18.3)
Insomnia Severity Index, *n* (%)		
no insomnia	2647	(57.0)
mild insomnia	1528	(32.9)
moderate or severe insomnia	469	(10.1)
*n* = 4664, Mdn = 6, M (SD)	7.3	(5.3)
Two PRIME-MD screening questions for depression, *n* (%)		
screen: positive	1534	(32.2)
screen: negative	3227	(67.8)
Has your work with confirmed or suspected cases of COVID-19 included exceptionally disturbing or distressing assignments? *n* (%)		
yes	609	(13.0)
no	4080	(87.0)
Have you had strong anxiety due to your own or close one’s risk of contracting serious illness due to your work with confirmed or suspected cases of? *n* (%)		
yes	934	(19.9)
no	3768	(80.1)
Have you or your close one contracted severe COVID-19 requiring hospital care? *n* (%)		
yes	134	(2.8)
no	4580	(97.2)
Has a close one to you died of COVID-19? *n* (%)		
yes	39	(0.8)
no	4687	(99.2)
Potentially traumatic events, PTEs, concerning working with COVID-19 patients, suspected COVID-19 cases, or contracting serious illness, *n* (%)		
at least one	1296	(27.8)
none	3358	(71.2)

**Table 3 ijerph-17-07905-t003:** Self-reported emotional distress and psychological symptoms among frontline personnel and other HUS personnel participants of the well-being survey in June 2020.

	**Were You in Contact with Confirmed or Suspected Cases of Last Week?**				
	**Directly Cared**		**Other Answers**		***p***
	***n***	**%**	***n***	**%**	
MHI-5					<0.001
>52	966	79.0	2997	84.8	
≤52	257	21.0	538	15.2	
ISI					<0.001
no insomnia	623	51.5	2016	58.9	
mild insomnia	438	36.2	1085	31.7	
moderate or severe insomnia	149	12.3	320	9.4	
PRIME-MD					0.030
screen: negative	796	65.3	2422	68.7	
screen: positive	423	34.7	1105	31.3	
PTEs: total (COVID-19-related)					<0.001
at least one reported yes	532	43.4	760	21.8	
none	693	56.6	2719	78.2	
PC-PTSD-5 (of those reporting at least one PTE)					0.832
screen: negative, <3	406	76.9	579	76.4	
screen: positive, ≥3	122	23.1	179	23.6	
OASIS (of those reporting at least one PTE)					0.410
screen: negative, < 8	386	72.6	534	70.4	
screen: positive, ≥ 8	146	27.4	224	29.6	
	**Did You Feel a Need for Psychological Support Last Month?**				
	**Yes**		**No**		
	***n***	**%**	***n***	**%**	***p***
MHI-5					<0.001
>52	362	47.0	3567	90.5	
≤52	408	53.0	373	9.5	
ISI					<0.001
no insomnia	223	29.3	2391	62.5	
mild insomnia	331	43.6	1176	30.8	
moderate or severe insomnia	206	27.1	256	6.7	
PHQ-2					<0.001
screen: negative	179	23.3	3013	76.6	
screen: positive	590	76.7	919	23.4	
PTEs: total (COVID-19 related)					<0.001
at least one reported yes	404	53.8	873	22.3	
none	347	46.2	3036	77.7	
PC-PTSD-5 (of those reporting at least one PTE)					<0.001
screen: negative, <3	229	56.8	742	85.4	
screen: positive, ≥3	174	43.2	127	14.6	
OASIS (of those reporting at least one PTE)					<0.001
screen: negative, <8	188	46.5	720	82.8	
screen: positive, ≥8	216	53.5	150	17.2	

**Table 4 ijerph-17-07905-t004:** Logistic regression models on the relation of sex, age, COVID-19 patient contact, potentially traumatic events (PTEs), and working as nurse and positive MHI-5 (N_models 1 & 3_ = 4672, N_model 2_ = 4531). (OR = odds ratio; CI = 95% confidence interval) ^1^.

Predictor	Model 1		Model 2		Model 3	
	OR	CI	OR	CI	OR	CI
(Intercept)	0.12	0.09–0.16	0.08	0.06–0.11	0.13	0.09–0.17
sex (woman)	1.60	1.20–2.13	1.49	1.10–2.02	1.59	1.19–2.11
age [40, 50]	0.83	0.68–1.01	0.91	0.74–1.12	0.83	0.68–1.01
age [50, 70]	0.62	0.51–0.76	0.69	0.56–0.85	0.62	0.51–0.76
age unknown	0.95	0.69–1.30	0.96	0.69–1.35	0.95	0.69–1.30
COVID-19 contact	1.23	1.03–1.47	0.93	0.77–1.13	0.70	0.39–1.27
nurse	1.40	1.17–1.67	1.14	0.94–1.38	1.30	1.08–1.58
PTEs	‒	‒	5.05	4.26–6.00	‒	‒
contact*nurse	‒	‒	‒	‒	1.88	1.01–3.50

^1^ The covariates (predictor variables) were binary-valued (0 or 1, reference age: 15–40). contact*nurse is an interaction term of COVID-19 patient contact and working as a nurse.

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
