# Peer review of "Personnel Well-Being in the Helsinki University Hospital during the COVID-19 Pandemic—A Prospective Cohort Study"

_ijerph, 2020, doi:10.3390/ijerph17217905_

Round 1
Reviewer 1 Report
This research paper presents an interesting topic of research relevant to the scope of this scientific journal. This research study is very important in providing the needed psychosocial support services to the front-line healthcare personnel who faced traumatic events during COVID-19 pandemic. This paper provides a good discussion of all obtained results. The level of novelty and creativity is considered good in this research study, and all results are new. However, this research paper suffers from major typos and grammatical errors at almost all parts. In order to consider this research paper for a possible publication in the International Journal of Environment Research and Public Health (IJERPH), this paper needs a major English language (writing) editing in all parts. Therefore, I would like to suggest my list of the recommended corrections for all parts that are needed to be edited in order to make it easy for authors to revise and edit their research paper shortly. Please note that the following sentences/words/phrases inside brackets are the suggested ones (corrected revision) by the referee:
* “On (Change “On” to “In”) March 2020 (Add comma “,”) strict measures took place in Finland to limit the COVID -19 pandemic.”
* “A majority of the Finnish COVID -19 –patients (shorten the distances between dash and words, and change it to “COVID-19 patients”) have been located in the southern Finland and consequently cared for in the HUS (Please describe what is the HUS so please change it to “Hospital District of Helsinki and Uusimaa (HUS)”) Helsinki University Hospital.”
* “The baseline survey in June 2020 was sent to 25494 HUS employees 25 out of whom 4804 (19%) answered; altogether 62.4% of the respondents were nursing staff and 8.9% 26 medical doctors.” (Change the whole sentence to: “In June 2020, the baseline survey was sent to 25494 HUS employees out of whom 4804 (19%) answered; altogether 62.4% of the respondents were nursing staff and 8.9% were medical doctors.”).
* “While the follow-up continues for a year and a half, this report shares the sociodemographic characteristics of the respondents and the first results of psychological symptoms from the (Change “the” to “our”) baseline survey.”
* “Out of those who were directly involved in pandemic patients` care, 43.4% reported potentially traumatic COVID-19 pandemic-related experiences vs. 21.8% among the other (Change “other” to “others”) (p < 0.001).”
* “While over a half of the personnel was symptomless, a group of respondents reported pandemic work –related (Please shorten the distance between dash and words, please change it to “work-related”) traumatic events and concurrent depressive, insomnia and anxiety symptoms.”
* “This highlights the need to ensure appropriate psychosocial support services to all traumatized personnel and PTEs (please state what is PTEs, so please change it to “potentially traumatic work-related events (PTEs)”) (Add “who”) were present especially among nursing staff.”
* “COVID-19 (severe acute respiratory syndrome coronavirus 2, SARS-CoV-2) outbreak began in December 2019 in Wuhan, China and has (Please delete “has”) caused a pandemic with over a 27.9 million confirmed infections and over 0.9 million deaths by September 10, 2020 [1].”
* “In most cases, the virus only causes (Change “only causes” to “causes only”) a mild disease.”
* “In Finland as globally (Change “as globally” to “from global perspective”), strict restrictions have taken place to slow down the pandemic by preventing physical contact between people.”
* “This is of utmost importance to secure intensive care (IC) capacity to those with severe symptoms who can benefit from it (Please remove this part “who can benefit from it”).”
* By September 14, 2020 there is detailed information on 336 deaths caused by COVID-19 pandemic in Finland, of them 48% were male and 52% female with a median age of 84 years.” (Change the whole sentence to: By September 14, 2020 (Add comma “,”) there is detailed information on 336 deaths caused by COVID-19 pandemic in Finland, of them 48% and 52% were males and females with a median age of 84 years, respectively.”).
* “Before they died of COVID-19 pandemic (Add comma “,” here) 35% were cared for in the primary health care, 20% in special medical care, 43% in social welfare services (Add comma “,”) and 2% at home or elsewhere (www.thl.fi/en) (Please change “(www.thl.fi/en)” to “(see also www.thl.fi/en)”).”
* “Health care personnel face unique challenges during (Please add “the COVID-19”) pandemic.”
* “In China, Wuhan area, frontline nurses and doctors caring for COVID-19 patients reported (Add “a” here) increase in symptoms of depression, insomnia, anxiety and psychological distress as compared to other health care personnel [4,5].”
* “Research on this field is limited and (Add here “according”) to our knowledge, only a few studies from Europe are currently available [8,10-11].”
* “The basic principles of high-quality psychosocial support [12-19] emerge (Change “emerge” to “have emerged”) from several international reports assessing the immediate needs of health care personnel (add here “who are”) caring for COVID-19 patients can shortly be summarized as follows; Listen, Supply, Prepare, Support and if needed - Care for us and our close ones [14-19].”
* “Timing is of importance in assessing stress -related (Change it to “stress-related”) symptoms, assessment before one-month duration from a potentially traumatic event is prone to wrong positive findings [20].”
* “COVID-19 pandemic increases the risk of exposure to potentially traumatic events among health care personnel in professional and private life, while pandemic in (remove “it”) itself is not always a traumatic event to everyone exposed to it [12-13].”
* “In this study, (Add here “the”) personnel well-being in the HUS Helsinki University Hospital during COVID-19 pandemic is followed via an electronic survey.”
* “We report baseline results from June 2020 from the prospective cohort study on HUS personnel’s psychological symptoms.” (Change the whole sentence to: “From June 2020, we report baseline results from the prospective cohort study on HUS personnel’s psychological symptoms.”).
* “In addition, the survey include (Change it to “includes”) questions about potential changes in respondents’ daily work and their adjustment to the changes, respondents’ attitudes towards COVID-19 patients, and (Change “, and” to “with”) a few open questions.”
* “Initially, all employees with a functional HUS email 85 address (N=25494) were invited to participate to (Change “to” to “in”) the baseline survey.”
* “Due to possible personnel work changes and turnover (Add comma “,” here) an open access link is also available in the HUS personnel’s internal website HUS Intra).”
* “A majority of answers was (Change it to “were”) received through email survey when launching the study during (Add here “the period from June 4, 2020 to 26, 2020”) but we (Add here “also”) included also (Delete “also” here) results from the open access link from the same time frame (Add comma “,” here) and (Add here “we”) also compared the answerers and report possible differences.”
* “We examined 2-way tables and Chi-squared tests in the former, and multivariate (multiple) logistic regression models in the latter to evaluate interaction effects of COVID-19 contact, potentially traumatic work -related (Change it to “work-related”) events (PTEs), and nursing-staff membership on the psychological outcomes, as well as to adjust the main effects for each other.”
* “Briefly, there was a statistically significant difference between first-line (Please change it to “front-line”) and other personnel in psychological distress (MHI-5), insomnia (ISI) and depressive symptoms (PRIME-MD).”
* “Studies from China have found that first line (Change it to “front-line) health care personnel, especially nursing staff, caring for patients with COVID-19 are at risk for anxiety and mental health problems [4, 31].”
* “Moreover, it has been identified that (Add here “the”) perceived threat of COVID-19 enhances turnover intentions among nurses [35].”
* “Also in this study especially nurses appeared to suffer a heavier psychological load from treating 181 COVID-19 patients than the other professionals.” (Change the whole sentence to: “In this study, nurses also appeared to suffer a heavier psychological load from treating COVID-19 patients than the other professionals.”)
* “To conclude, also in Finland in addition to first-line personnel, especially nursing staff all personnel who report potentially traumatic events related to COVID-19 pandemic require attention and support.” (Change the whole sentence to: “In Finland, in addition to front-line personnel, especially nursing staff, all personnel who report potentially traumatic events related to COVID-19 pandemic require attention and support.”).
In addition, I have the following recommended changes:
* Please change “first-line personnel” in the whole paper to “front-line personnel”.
* Please add a conclusion section at the end of this paper. You can take the last part of the discussion section in this paper and add it to a separate conclusion section.
In conclusion, editing all those recommended corrections can improve the quality of this research paper. This research paper has enough new original results that match the scope of this prestigious journal.
Author Response
* “On (Change “On” to “In”) March 2020 (Add comma “,”) strict measures took place in Finland to limit the COVID -19 pandemic.”
We are grateful for the time reviewer 1 has used to comment this ms and the comments helped us to clarify the ms. The English language has thoroughly been assessed and suggested corrections made. Due to word restriction we shortened the ending in the Abstract a bit more. Please see below detailed comments
“On (Change “On” to “In”) March 2020 (Add comma “,”) strict measures took place in Finland to limit the COVID -19 pandemic.” Response: Corrected as suggested.
* “A majority of the Finnish COVID -19 –patients (shorten the distances between dash and words, and change it to “COVID-19 patients”) have been located in the southern Finland and consequently cared for in the HUS (Please describe what is the HUS so please change it to “Hospital District of Helsinki and Uusimaa (HUS)”) Helsinki University Hospital.” Response: Corrected as suggested.
* “The baseline survey in June 2020 was sent to 25494 HUS employees 25 out of whom 4804 (19%) answered; altogether 62.4% of the respondents were nursing staff and 8.9% 26 medical doctors.” (Change the whole sentence to: “In June 2020, the baseline survey was sent to 25494 HUS employees out of whom 4804 (19%) answered; altogether 62.4% of the respondents were nursing staff and 8.9% were medical doctors.”). Response: Corrected as suggested.
* “While the follow-up continues for a year and a half, this report shares the sociodemographic characteristics of the respondents and the first results of psychological symptoms from the (Change “the” to “our”) baseline survey.”Response: Corrected as suggested.
* “Out of those who were directly involved in pandemic patients` care, 43.4% reported potentially traumatic COVID-19 pandemic-related experiences vs. 21.8% among the other (Change “other” to “others”) (p < 0.001).”
Response: Corrected as suggested, and comma corrected
* “While over a half of the personnel was symptomless, a group of respondents reported pandemic work –related (Please shorten the distance between dash and words, please change it to “work-related”) traumatic events and concurrent depressive, insomnia and anxiety symptoms.” Response: Sentence was rephrased.
* “This highlights the need to ensure appropriate psychosocial support services to all traumatized personnel and PTEs (please state what is PTEs, so please change it to “potentially traumatic work-related events (PTEs)”) (Add “who”) were present especially among nursing staff.” Response: Sentence rephrased; "This highlights the need to ensure appropriate psychosocial support services to all traumatized personnel and especially nursing staff may require attention."
* “COVID-19 (severe acute respiratory syndrome coronavirus 2, SARS-CoV-2) outbreak began in December 2019 in Wuhan, China and has (Please delete “has”) caused a pandemic with over a 27.9 million confirmed infections and over 0.9 million deaths by September 10, 2020 [1].”Response: Corrected as suggested.
* “In most cases, the virus only causes (Change “only causes” to “causes only”) a mild disease.”Response: Corrected as suggested.
* “In Finland as globally (Change “as globally” to “from global perspective”), strict restrictions have taken place to slow down the pandemic by preventing physical contact between people.” Response: Corrected as suggested.
* “This is of utmost importance to secure intensive care (IC) capacity to those with severe symptoms who can benefit from it (Please remove this part “who can benefit from it”).” Response: Corrected as suggested.
* By September 14, 2020 there is detailed information on 336 deaths caused by COVID-19 pandemic in Finland, of them 48% were male and 52% female with a median age of 84 years.” (Change the whole sentence to: By September 14, 2020 (Add comma “,”) there is detailed information on 336 deaths caused by COVID-19 pandemic in Finland, of them 48% and 52% were males and females with a median age of 84 years, respectively.”). Response: Corrected as suggested. The information was updated responding to the current situation.
* “Before they died of COVID-19 pandemic (Add comma “,” here) 35% were cared for in the primary health care, 20% in special medical care, 43% in social welfare services (Add comma “,”) and 2% at home or elsewhere (www.thl.fi/en) (Please change “(www.thl.fi/en)” to “(see also www.thl.fi/en)”).”Response: Corrected as suggested, and “pandemic” omitted.
* “Health care personnel face unique challenges during (Please add “the COVID-19”) pandemic.” Response: Corrected as suggested.
* “In China, Wuhan area, frontline nurses and doctors caring for COVID-19 patients reported (Add “a” here) increase in symptoms of depression, insomnia, anxiety and psychological distress as compared to other health care personnel [4,5].”Response: Added “an” increase
* “Research on this field is limited and (Add here “according”) to our knowledge, only a few studies from Europe are currently available [8,10-11].”Response: Corrected as suggested.
* “The basic principles of high-quality psychosocial support [12-19] emerge (Change “emerge” to “have emerged”) from several international reports assessing the immediate needs of health care personnel (add here “who are”) caring for COVID-19 patients can shortly be summarized as follows; Listen, Supply, Prepare, Support and if needed - Care for us and our close ones [14-19].”
Response: Corrected as suggested.
* “Timing is of importance in assessing stress -related (Change it to “stress-related”) symptoms, assessment before one-month duration from a potentially traumatic event is prone to wrong positive findings [20].” Response: Corrected as suggested.
* “COVID-19 pandemic increases the risk of exposure to potentially traumatic events among health care personnel in professional and private life, while pandemic in (remove “it”) itself is not always a traumatic event to everyone exposed to it [12-13].” Response: “in” removed
* “In this study, (Add here “the”) personnel well-being in the HUS Helsinki University Hospital during COVID-19 pandemic is followed via an electronic survey.”Response: Corrected as suggested.
* “We report baseline results from June 2020 from the prospective cohort study on HUS personnel’s psychological symptoms.” (Change the whole sentence to: “From June 2020, we report baseline results from the prospective cohort study on HUS personnel’s psychological symptoms.”).
Response: Corrected as suggested.
* “In addition, the survey include (Change it to “includes”) questions about potential changes in respondents’ daily work and their adjustment to the changes, respondents’ attitudes towards COVID-19 patients, and (Change “, and” to “with”) a few open questions.”
Response: Corrected and rephrased.
* “Initially, all employees with a functional HUS email 85 address (N=25494) were invited to participate to (Change “to” to “in”) the baseline survey.” Response: Corrected as suggested.
* “Due to possible personnel work changes and turnover (Add comma “,” here) an open access link is also available in the HUS personnel’s internal website HUS Intra).” Response: Corrected as suggested.
* “A majority of answers was (Change it to “were”) received through email survey when launching the study during (Add here “the period from June 4, 2020 to 26, 2020”) but we (Add here “also”) included also (Delete “also” here) results from the open access link from the same time frame (Add comma “,” here) and (Add here “we”) also compared the answerers and report possible differences.” Response: Corrected as suggested.
* “We examined 2-way tables and Chi-squared tests in the former, and multivariate (multiple) logistic regression models in the latter to evaluate interaction effects of COVID-19 contact, potentially traumatic work -related (Change it to “work-related”) events (PTEs), and nursing-staff membership on the psychological outcomes, as well as to adjust the main effects for each other.”
Response: Corrected as suggested.
* “Briefly, there was a statistically significant difference between first-line (Please change it to “front-line”) and other personnel in psychological distress (MHI-5), insomnia (ISI) and depressive symptoms (PRIME-MD).” Response: Corrected as suggested.
* “Studies from China have found that first line (Change it to “front-line) health care personnel, especially nursing staff, caring for patients with COVID-19 are at risk for anxiety and mental health problems [4, 31].”Response: Corrected as suggested.
* “Moreover, it has been identified that (Add here “the”) perceived threat of COVID-19 enhances turnover intentions among nurses [35].”Response: Corrected as suggested.
* “Also in this study especially nurses appeared to suffer a heavier psychological load from treating 181 COVID-19 patients than the other professionals.” (Change the whole sentence to: “In this study, nurses also appeared to suffer a heavier psychological load from treating COVID-19 patients than the other professionals.”)
Response: Corrected as suggested.
* “To conclude, also in Finland in addition to first-line personnel, especially nursing staff all personnel who report potentially traumatic events related to COVID-19 pandemic require attention and support.” (Change the whole sentence to: “In Finland, in addition to front-line personnel, especially nursing staff, all personnel who report potentially traumatic events related to COVID-19 pandemic require attention and support.”). Response: Corrected as suggested.
In addition, I have the following recommended changes:
* Please change “first-line personnel” in the whole paper to “front-line personnel”.
Response: Corrected as suggested.
* Please add a conclusion section at the end of this paper. You can take the last part of the discussion section in this paper and add it to a separate conclusion section. Response: Separate conclusion section was added.
Reviewer 2 Report
Important contribution to the special mental health burden of health workers in the context of the Covid 19 pandemic
Comments/questions/recommendations
Form: in the text are a whole series of formal errors, which are mainly due to hyphen formulations (e.g. “Covid -19 – patients” Line 22, “work –related”, Line 32 etc), spacing between characters (e.g. "36.4%;n=1011" Line 102) or comma settings (e.g. "23-4%", Line 108). Please check the whole text again
Introduction, Line 38-40: here the numbers could be updated again, e.g. 38 Mio confirmed infections and about 1,1 Mio deaths by October 14, 2020
Table headings (for the editor): should be unified
Table 2 (for the editor): partially too large line spacing; second last line is 27.5 bold
Table 2 and or 3: For the scales MHI-5, ISI and PHQ-2, the mean value and standard deviation of the total group and the respective classes (e.g. <52/<= 52) could also be given
Table 4: header “nModels 1 & 3” and “nModel2”: "models" should be in subscript; also here is a different formatting (font sizes, line spacing etc. compared to table 3)
Results/Line 115-119: the results of the model estimates or logistic regression analyses could be described in more detail
Discussion/Line 149ff: some prevalence estimates of the psychosocial burden on nurses or other health professionals with regard to mental health/psychological disorders (especially depression, anxiety disorders, post-traumatic stress disorders) could be used for comparison; e.g. [1–7]
Discussion/Line 184ff: the demand for stronger psychosocial or psychotherapeutic support, especially for nurses, could be discussed in more detail, especially with regard to what acutely stressed nurses can be offered locally and which support possibilities already exist and can be used, e.g. [8–10]
Translated by www.DeepL.com/Translator (free version)
Literatur
1 Gupta S, Sahoo S. Pandemic and mental health of the front-line healthcare workers. A review and implications in the Indian context amidst COVID-19. Gen Psych 2020; 33 (5): e100284; DOI: 10.1136/gpsych-2020-100284
2 Muller AE, Hafstad EV, Himmels JPW et al. The mental health impact of the covid-19 pandemic on healthcare workers, and interventions to help them: A rapid systematic review. Psychiatry Res 2020; 293: 113441; DOI: 10.1016/j.psychres.2020.113441
3 Shechter A, Diaz F, Moise N et al. Psychological distress, coping behaviors, and preferences for support among New York healthcare workers during the COVID-19 pandemic. Gen Hosp Psychiatry 2020; 66: 1–8; DOI: 10.1016/j.genhosppsych.2020.06.007
4 Pappa S, Ntella V, Giannakas T et al. Prevalence of depression, anxiety, and insomnia among healthcare workers during the COVID-19 pandemic. A systematic review and meta-analysis. Brain, Behavior, and Immunity 2020; 88: 901–907; DOI: 10.1016/j.bbi.2020.05.026
5 Mo Y, Deng L, Zhang L et al. Work stress among Chinese nurses to support Wuhan in fighting against COVID‐19 epidemic. J Nurs Manag 2020; 28 (5): 1002–1009; DOI: 10.1111/jonm.13014
6 Liang Y, Chen M, Zheng X et al. Screening for Chinese medical staff mental health by SDS and SAS during the outbreak of COVID-19. Journal of Psychosomatic Research 2020; 133: 110102; DOI: 10.1016/j.jpsychores.2020.110102
7 Spoorthy MS, Pratapa SK, Mahant S. Mental health problems faced by healthcare workers due to the COVID-19 pandemic–A review. Asian Journal of Psychiatry 2020; 51: 102119; DOI: 10.1016/j.ajp.2020.102119
8 Buselli R, Baldanzi S, Corsi M et al. Psychological Care of Health Workers during the COVID-19 Outbreak in Italy. Preliminary Report of an Occupational Health Department (AOUP) Responsible for Monitoring Hospital Staff Condition. Sustainability 2020; 12 (12): 5039; DOI: 10.3390/su12125039
9 Duan L, Zhu G. Psychological interventions for people affected by the COVID-19 epidemic. The Lancet Psychiatry 2020; 7 (4): 300–302; DOI: 10.1016/S2215-0366(20)30073-0
10 Maben 1 2J, Bridges J. Covid‐19. Supporting nurses' psychological and mental health. J Clin Nurs 2020; 29 (15-16): 2742–2750; DOI: 10.1111/jocn.15307
Author Response
We are grateful for the time reviewer 2 has used to this ms and the comments were helpful. For detailed comments, please see below.
Form: in the text are a whole series of formal errors, which are mainly due to hyphen formulations (e.g. “Covid -19 – patients” Line 22, “work –related”, Line 32 etc), spacing between characters (e.g. "36.4%;n=1011" Line 102) or comma settings (e.g. "23-4%", Line 108). Please check the whole text again
Response: The whole text was revisited and checked for these errors.
Introduction, Line 38-40: here the numbers could be updated again, e.g. 38 Mio confirmed infections and about 1,1 Mio deaths by October 14, 2020
Response: The numbers were updated.
Table headings (for the editor): should be unified
Response: We have unified the headings and their visual format.
Table 2 (for the editor): partially too large line spacing; second last line is 27.5 bold
Table 2 and or 3: For the scales MHI-5, ISI and PHQ-2, the mean value and standard deviation of the total group and the respective classes (e.g. <52/<= 52) could also be given.
Response: We provide mean value and standard deviation for MHI-5 and ISI in the table.
Table 4: header “nModels 1 & 3” and “nModel2”: "models" should be in subscript; also here is a different formatting (font sizes, line spacing etc. compared to table 3)
Response: We have changed the Table 4 formatting as requested (though the journal presumably imposes its own formatting later on, if accepting the MS).
Results/Line 115-119: the results of the model estimates or logistic regression analyses could be described in more detail
Response: We have described the results from Table 4 in more detail as requested (lines 122-128):
“Table 4 reveals that potentially traumatic COVID-19 pandemic related events strongly predicted psychological distress indexed by MHI-5 (Model 2). Both belonging to nursing staff and participating to direct care of Covid-19 patients were independently associated with psychological distress (Model 1, Table 4) and these associations were explained by (i.e., lost significance upon adjusting for) experiencing Covid-19 pandemic related traumatic events (Model 2). Furthermore, participation to direct care of Covid-19 patients was not associated with psychological distress in non-nursing staff (Model 3), though relatively few non-nurses participated (n = 136 vs. 1094).”
Discussion/Line 149ff: some prevalence estimates of the psychosocial burden on nurses or other health professionals with regard to mental health/psychological disorders (especially depression, anxiety disorders, post-traumatic stress disorders) could be used for comparison; e.g. [1–7]
Response: Comparison to earlier literature was added to discussion, as suggested.
Discussion/Line 184ff: the demand for stronger psychosocial or psychotherapeutic support, especially for nurses, could be discussed in more detail, especially with regard to what acutely stressed nurses can be offered locally and which support possibilities already exist and can be used, e.g. [8–10]
Response: We have now a sentence "There is a demand for stronger psychosocial or psychotherapeutic support, especially for nurses, and already existing support possibilities such as peer and team support could be used" before Conclusions and literature additions were also made.
We have also thoroughly updated formatting of references, additions are shown with color.